# Modeling di (2-ethylhexyl) Phthalate (DEHP) and Its Metabolism in a Body’s Organs and Tissues through Different Intake Pathways into Human Body

**DOI:** 10.3390/ijerph19095742

**Published:** 2022-05-09

**Authors:** Ao Li, Lingyi Kang, Runjie Li, Sijing Wu, Ke Liu, Xinke Wang

**Affiliations:** School of Human Settlements and Civil Engineering, Xi’an Jiaotong University, Xi’an 710049, China; kichelle620@163.com (A.L.); kanglingyi123@163.com (L.K.); lirunjie@stu.xjtu.edu.cn (R.L.); wsj13572050386@163.com (S.W.); mjyrke95@163.com (K.L.)

**Keywords:** exposure, indoor, metabolism, PBPK model, phthalate esters

## Abstract

Phthalate esters (PAEs) are ubiquitous in indoor environments as plasticizers in indoor products. Residences are often exposed to indoor PAEs in the form of gas, particles, settled dust, and surface phases. To reveal the mechanism behind the accumulation of PAEs in different tissues or organs such as the liver and the lungs when a person exposed to indoor PAEs with different phases, a whole-body physiologically based pharmacokinetic model for PAEs is employed to characterize the dynamic process of phthalates by different intake pathways, including oral digestion, dermal adsorption, and inhalation. Among three different intake pathways, dermal penetration distributed the greatest accumulation of DEHP in most of the organs, while the accumulative concentration through oral ingestion was an order of magnitude lower than the other two doses. Based on the estimated parameters, the variation of di-ethylhexyl phthalate (DEHP) and mono (2-ethylhexyl) phthalate (MEHP) concentration in the venous blood, urine, the liver, the thymus, the pancreas, the spleen, the lungs, the brain, the heart, and the kidney for different intake scenarios was simulated. The simulated results showed a different accumulation profile of DEHP and MEHP in different organs and tissues and demonstrated that the different intake pathways will result in different accumulation distributions of DEHP and MEHP in organs and tissues and may lead to different detrimental health outcomes.

## 1. Introduction

Phthalate esters (PAEs), a kind of semi-volatile organic compound, have been widely used as plasticizers in household and industrial products for decades, primarily to increase the flexibility and resilience of polyvinyl chloride (PVC) products. Since they are not chemically bounded to a product matrix, PAEs will be gradually released from indoor products into the air and tend to adhere to suspended particulate matter, dust, and indoor surfaces because of their low vapor pressure. Thus, PAEs have been detected in almost all indoor environments [1,2,3,4,5,6], and exposure to indoor PAEs seems especially severe in China [7,8]. What is more, during the on-going COVID-19 pandemic, people have been wearing masks for a long period of time during their daily lives, which may have increased exposure to a certain number of phthalates of an average level of 1950 ng/g, determined in mask samples [9].

Epidemiological and toxicological studies have shown that exposure to PAEs through indoor sources may be related to carcinogenesis, the disruption of the endocrine system, and other symptoms [2]. As a kind of endocrine disruption, PAEs have an impact on the endocrine system and the genital system [10]. Animal [11], as well as human [12], sampling tests indicated that PAEs will cause damage to the reproductive system and the testicular function of males. In addition, epidemiology investigations suggested that early puberty in females is related to DEHP exposure [13,14], which may also raise the risk of breast cancer and endometriosis [15]. Besides the adverse effects on the endocrine system and the reproductive system, PAEs may even cause allergies, asthma [16], and cancer [17,18].

PAEs enter the human body through three kinds of intake ways: inhalation, oral ingestion, and dermal penetration [19]. The mechanism of emission of PAEs from sources and their transport fate indoors have been basically clarified by many studies [20,21,22] so that the indoor concentration of PAEs can be predicted or estimated with accepted accuracy by the models, together with numerous measured values in different countries and regions [1,23]. At the same time, the indoor concentration of phthalate ester and its metabolism in urine and blood in different countries and regions has also been measured for health risk assessments [24,25,26]. However, the relationship between indoor exposure to phthalate and metabolisms has been not established, and the prediction of internal exposure based on the external exposure to PAEs is difficult. Recently, some researchers have attempted to explore the issue of how PAEs entered human bodies based on some PBPK (physiologically based pharmacokinetic) models cooperated with intakes of dermal penetration and inhalation, as well as some experimental data [27,28,29]. However, the detrimental health outcomes of different intake pathways still cannot be compared in the mechanism. In addition, for linking external exposure to DEHP with internal exposure, the distribution of PAEs in the body’s organs and tissues after entering the human body through three pathways, including inhalation, oral ingestion, and dermal penetration, still needs to be investigated. What is more, some important organs and tissues are not specified in those models, so the internal exposure to PAEs and their substitutes in these organs and tissues cannot be estimated. Therefore, the objective of this paper is to made a preliminary study on this issue by using a whole-body PBPK model. This PBPK model can simulate targeted organs and tissues through different intake pathways and therefore can be beneficial to study how the accumulation of PAEs leads to detrimental health outcomes in different organs in the human body.

## 2. Materials and Methods

### 2.1. Model Description

Generally, physiologically based pharmacokinetic (PBPK) models are used for decisions in drug invention and developments and can simulate simultaneous concentrations of drugs with time in different organs and tissues after a specified dosing. Recently, some studies have employed PBPK models to investigate contaminants’ exposure and their metabolism variation with time after exposure [29]. In this paper, we referred to the generic 14-compartments PBPK model in the Simbiology platform in Matlab [30]. As studies have shown that DEHP might have the adverse effects on carcinogenesis, the disruption of the endocrine system, asthma, and other symptoms [16,31], the simulation chose the liver, the lungs, and the endocrine system (the spleen and the pancreas) as target organs to simulate their accumulation time course. The original model assumes that drugs are well stirred and absorption rates are determined merely by transcellular or paracellular permeability and solubility to access the accuracy of the prediction. However, this model cannot be applied well to predict the concentration-time profiles of PAEs in human organs in three different ways (inhalation, oral ingestion, and dermal penetration). So as shown in Figure 1, we divided the original model into numerous compartments: the lungs, the heart, the brain, the muscles, the skin, the liver, the gut, the spleen, the pancreas, the bones, the kidneys, urine, the thymus, arterial blood, venous blood, and the rest of the organs. The blue lines represent the DEHP flow between organs, and the red ones represents the MEHP towards the organs. Among the organs, urine, which represents the excretion of the human body, was regarded as a compartment with a volume of 1 milligram. Additionally, the compartment representing the rest of organs only presented the metabolism and delivery of MEHP as the hydrolysis product of DEHP. DEHP entered the body through the gut, the lungs, and the skin, representing the three pathways of ingestion intake, inhalation, and dermal penetration, respectively.

The reactions between the species in the PBPK model were referred to in Sharma’s modeling approach’s modeling approach [27], which mainly discussed 5 metabolites: MEHP, mono (2-ethyl-5-hydroxyhexyl) phthalate (5-OH MEHP), mono (2-ethyl-5-oxohexyl) phthalate (5oxo-MEHP), mono (2-ethyl-5-carboxypentyl) phthalate (5cx MEPP), and phthalic acid esters. It was only kinetic in the intestines and the liver [25], and only MEHP was concerned with simplification. The main governing equations are shown in Appendix A.

### 2.2. Model Parameter

As shown in Table 1, the parameters of the chemical and physical properties of DEHP and MEHP in the reactions in the PBPK model were referred to in Sharma’s modeling approach [27]. The basic human body parameters such as organ capacity in Table 2 refer to the original PBPK model [30]. Due to the lack of a link between the parameters and MEHP delivery, as well as the metabolism in the original model built by Peters, more related parameters were referred to Sharma and applied in the improved model. According to Sharma’s research, the model parameters are distributed log normally in the range of ±1 to ±1.5 standard deviations. By estimating uncertainty, the value was picked by selecting a random value first and ran by the Monte Carlo simulations. After 20,000 iterations, the collected output values formed a random sample. However, in this model, the parameters simplify chose the mean value.

### 2.3. Dose in PBPK Model

In this paper, DEHP was chosen as the target compound due to its ubiquitous existence in indoor environments. Three different dosing scenarios, as shown in Figure 2, were adopted to represent the process of DEHP entering the human body through inhalation, oral ingestion, and dermal penetration. For dermal penetration, pertinent studies showed that for those small-molecular-weight PAEs, dermal penetration contributes to a dominant intake compared to inhalation [37,38]. Additionally, a later study also proved that, compared to inhalation, skin exposure has a greater impact on the human body through a more detailed model and experimental studies [19,39]. For inhalation, due to the filtration of nasal mucosa, some of the chemical compounds can be effectively blocked. For oral ingestion, the model assumed that the intake amount of DEHP is correlated to the intake amount of food.

Based on the studies above and a field investigation of health exposure to DEHP, we set a new dosing amount of three kinds of intake.

According to the estimated data from studies [40,41], dermal penetration and inhalation were both set as continuous doses with amounts of 0.68 mg/h and 0.53 mg/h, respectively, because the human body was assumed to be exposed to an indoor contaminated environment all day. According to the research [42], the mean dietary intake of DEHP in the general population was 2.34 μg/kg/day, and 97.5% of the intake in the general population was 5.22 μg/kg/day. Therefore, the total intake amount of oral ingestion was set as 1.95 mg (3.25 μg/kg/day, 60 kg). The oral ingestion was assumed as 3 times with dinners at 8:00 am, 12:00 pm, and 18:00 pm, respectively, in one day (the duration of one intake lasts for 30 min). Meanwhile, based on the average amount of food intake of a person, the ratio for breakfast, lunch, and supper is 0.5:0.8:0.65 with regard to the different amounts of food. Other oral ingestion was neglected for the purposes of simplification.

## 3. Results and Discussion

### 3.1. Validation of PBPK Model

The validation of the new PBPK model was tested by the experimental data of metabolites of DEHP in the human urine and serum after single oral doses of deuterium-labelled DEHP [36]. According to Koch, the single dose was 48.1 mg (0.65 mg/kg body-weight, a male volunteer weighting 75 kg). We set the oral dose of the same amount in the PBPK model, and the time profile of the absolute accumulative excretion of MEHP in urine was shown in Figure 3. According to the figure, the MEHP accumulation (mg) in the urine of the simulation and of the experiment ascended with a similar trend and gradually reached an equilibrium at the amounts of 3.000 mg and 2.475 mg, respectively. Figure 4 shows that the trend and the peak of the MEHP concentration in the plasma between the simulation of the model and of the reference coincided, which suggests that the result validated the rationality of the model. According to figures, the MEHP amount of the excretion, as well as the concentration in plasma of the simulation, were comparatively higher than those of the experiment. This may indicate that the simulated human body in the model has greater metabolic capacity than the volunteer in Koch experiment. Another potential reason for the bias is the uncertainty of the estimated parameters in the model. This needs to be studied further in the future to estimate the parameters more accurately.

### 3.2. DEHP and MEHP Concentration in Different Organs and Tissues through Three Sifferent Intake Pathways

According to Figure 5, a 24-h dose could not be metabolized and excreted adequately until 48 h. This finding suggests that the leftover of DEHP the day before would affect the DEHP accumulative concentration the next day, which, on the other hand, could be out of consideration at the third day. Shown in Table 3, the error between 24 h and 48 h was in the range of 1.7924% to 2.004%, which cannot be ignored, while the difference between 48 h and 72 h was 0.0979% to 0.0674%, and the following error was getting smaller gradually. Regarding the human intakes of different kinds of PAEs in their daily lives, to study the actual effect of accumulative DEHP and MEHP in the human body, simulation is needed for at least 48 h.

#### 3.2.1. Oral Ingestion

Figure 6 shows that DEHP quickly accumulated in the organs after entering the human body through scheduled oral doses (0.5 mg at 8 h, 0.8 mg at 12 h, 0.65 mg at 18 h, 0.5 mg at 32 h, 0.8 mg at 36 h, and 0.65 mg at 42 h). Among the organs, the DEHP concentration accumulated most in the liver and reached a maximum value (2.608 × 10^−3^ µg/mL at 33 h, 4.180 × 10^−3^ µg/mL at 37 h, and 3.404 × 10^−3^ µg/mL at 43 h) within an hour after the intakes (at this time, each organ rapidly reached the cumulative peak). The highest DEHP concentration in the lungs, the spleen, and the pancreas (ranging from 9.221 × 10^−4^ µg/mL to 9.677 × 10^−4^ µg/mL at 33h, 1.512 × 10^−3^ µg/mL to 1.584 × 10^−3^ µg/mL at 37 h, and 1.275 × 10^−3^ µg/mL to 1.333 × 10^−3^ µg/mL at 43 h, respectively) was basically a half lower than the DEHP concentration in the liver, indicating that the intake of oral ingestion has the greatest adverse effect on liver. After reaching the peak, the DEHP in various organs quickly metabolized. As shown in the figure, the concentration of DEHP dramatically reduced in a short period of time. As the scheduled oral doses are constantly absorbed, the fewest cumulative concentration of DEHP in human body still remained in the range of 10^−5^ µg/mL to 10^−6^ µg/mL.

Figure 7 indicates the change of MEHP concentration in the organs though oral ingestion. Compared to the organs of the DEHP concentration simulation, the spleen and pancreas combined to form a whole endocrine system. As the hydrolysis product of DEHP, MEHP is gradually generated during the hydrolysis and metabolites at the same time, causing a trend of a constant increase in the first hour and then a reduction at a steady rate. According to Figure 7 compared to the time course of DEHP concentration, the curves of MEHP concentration in different organs showed the same order of the accumulative amount. The greatest distribution of the MEHP concentration was still in the liver (2.862 × 10^−3^ µg/mL at 33 h, 4.669 × 10^−3^ µg/mL at 37 h, and 3.759 × 10^−3^ µg/mL at 43 h, respectively), which was nearly two times as many as the accumulation zenith in the lungs and the endocrine system.

#### 3.2.2. Dermal Penetration

In the model, the dose through dermal penetration was continuously supplied. As a result, the DEHP concentration through skin rises gradually at first and then reaches equilibrium at a later stage. As the penetration through skin was a continuous dose, the accumulative concentration of DEHP through dermal penetration was, on average, an order of magnitude higher than the oral ingestion.

Figure 8 shows that the DEHP concentration among organs though dermal penetration increased at a steady rate until it reached a peak within 30 h and then reached an equilibrium concentration. The greatest equilibrium accumulation through dermal penetration was in the spleen and the pancreas (4.131 × 10^−2^ µg/mL). However, the DEHP concentration in the liver was the lowest among the organs (7.316 × 10^−3^ µg/mL), indicating that the dose through the skin may have had less of an impact on liver. What is more, the equilibrium concentration of the lungs (4.004 × 10^−2^ µg/mL) was slightly lower than that in the spleen and the pancreas.

Figure 9 shows the time-concentration profile of MEHP in various organs through dermal penetration. Compared to the curves of DEHP concentration in different organs, the time profile of MEHP showed a different order of the accumulative number of organs. The MEHP concentration in the liver (6.798 × 10^−3^ µg/mL) was markedly higher than the concentration in the endocrine system (4.383 × 10^−3^ µg/mL) and the lungs (4.248 × 10^−3^ µg/mL). This phenomenon can be explained by the fact that the DEHP in the liver was mostly metabolized and decomposed to MEHP and its other hydrolysates. In the MEHP simulation, the concentration of liver was still the greatest; however, the accumulation in the lungs and the endocrine system cannot be ignored. This simulation further proves that phthalates exposure has great potential adverse detrimental health outcomes on the liver.

#### 3.2.3. Inhalation

As a continuous dose, the method of inhalation distributes the lower intake amount of DEHP than the dermal penetration. The trend of DEHP concentration was similar to the dermal penetration but reached an equilibrium more rapidly, suggesting that DEHP in the lungs might transfer and decompose faster. According to the Figure 10 the descending order of the equilibrium peak of DEHP was the spleen and pancreas (3.221 × 10^−2^ µg/mL), the lungs (3.122 × 10^−3^ µg/mL), and the liver (5.703 × 10^−3^ µg/mL).

According to Figure 11 the MEHP concentration through inhalation shared a similar rising trend but a disparity in the order of the equilibrium amount in the organs. Among the organs, the liver still had the highest MEHP equilibrium concentration (5.299 × 10^−3^ µg/mL). However, the difference between the lungs (3,311 × 10^−3^ µg/mL) and the endocrine system (3.416 × 10^−3^ µg/mL) was small. As the liver and the gut are the only organs that can metabolize in the PBPK model, this result further proved that the DEHP in the liver was mostly metabolized and hydrolyzed to MEHP and other small molecule compounds.

### 3.3. Further Comparison and Health Care Assessment through Three Different Doses

The simulation showed that the constant and scheduled intake of phthalates cannot be metabolized and excreted entirely. The majority of DEHP can be metabolized in the first 48 h after intake. Few compounds (DEHP and its metabolites) remain in the body, and those parts of the compounds will eventually lead to potential detrimental health outcomes. This simulation result can be proved by Anderson et al., who suggest that the metabolites of DEHP excreted 47.1 ± 8.5% on a molar basis within 48 h [33]. Through the simulation, the intake of dermal penetration, as a repeated dose with a greater amount, distributes the greatest accumulation in most of the organs, while the oral ingestion, ignoring the breathing behavior via mouth, only intakes DEHP through dinners and therefore accumulates medially an order of magnitude lower than other two doses.

Studies have proved that DEHP and its hydrolysis product, MEHP, can do damage to human organs. According to Figure 12 and Figure 13, among all the selected organs, the lung, the pancreas, and the spleen accumulate more DEHP, while the liver accumulates smaller molecule compounds, MEHP. This result shows the metabolic function of the liver and further suggests the non-negligible accumulation of PAEs in different organs. The link with actual pathological studies, and phthalates accumulation in the liver, might have an impact on the pathways of some essential receptors in the liver that control not only xenobiotic detoxification but also energy homeostasis and the circadian clock [43]. The excessive accumulation of DEHP in liver might inhibit liver detoxifying enzymes and result in liver dysfunction [44].

What is more, DEHP might cause significant decreases in cell viability and impaired antioxidant systems [45], which might cause endocrine system disorders. Research also showed that particularly low molecular weight phthalates were associated with poorer lung function [46]. After all, the simulation suggests that the DEHP might have toxicity to human lung epithelial A549 cells and eventually cause DNA damage, oxidative stress, apoptosis, and estrogenic effects [47].

## 4. Conclusions

From the results, the proposed whole-body PBPK model is able to predict the internal exposure of DEHP and its metabolites through three pathways into the body: inhalation, oral ingestion, and dermal penetration. Through the simulation, among three doses, the intake of dermal penetration distributes the greatest accumulation in most of the organs, indicating that the intake through the skin may be the main intake method of daily exposure. The intake of dermal penetration and inhalation, as a continuous dose, leads to the equilibrium concentration of DEHP and MEHP in different organs and tissues (ranging from 10^−2^ to 10^−3^ µg/mL). While oral ingestion, distributing the lowest accumulation through the three kinds of intake pathways, retains much less concentration at the end of every day (ranging from 10^−5^ to 10^−6^ µg/mL). Additionally, the simulation result demonstrates that the endocrine system and the lung accumulate more DEHP, but the liver, as the main metabolizing organ, accumulate more small molecule compounds, MEHP. Certainly, the model still has potential. The simulation result that the DEHP concentration in the lungs through inhalation was less than the dermal penetration still needs to be discussed. We suggest that the amount of DEHP will decrease through transportation due to the adhesion or other mechanisms. However, the model assumes arterial blood and venous blood as a single compartment, respectively. This flaw lacks detailed reactions to account for the inevitable loss while the drug is flowing through vessels. What is more, as the physical and chemical properties of the chemical compounds can be adjusted, the PBPK model was believed to be compatible with other kinds of doses. To encourage reactions of various molecule compounds, the model can also be simulated with a greater framework of detailed profiles in the human body. This PBPK model can be beneficial if linked with pathology and environmental science.

## Figures and Tables

**Figure 1 ijerph-19-05742-f001:**
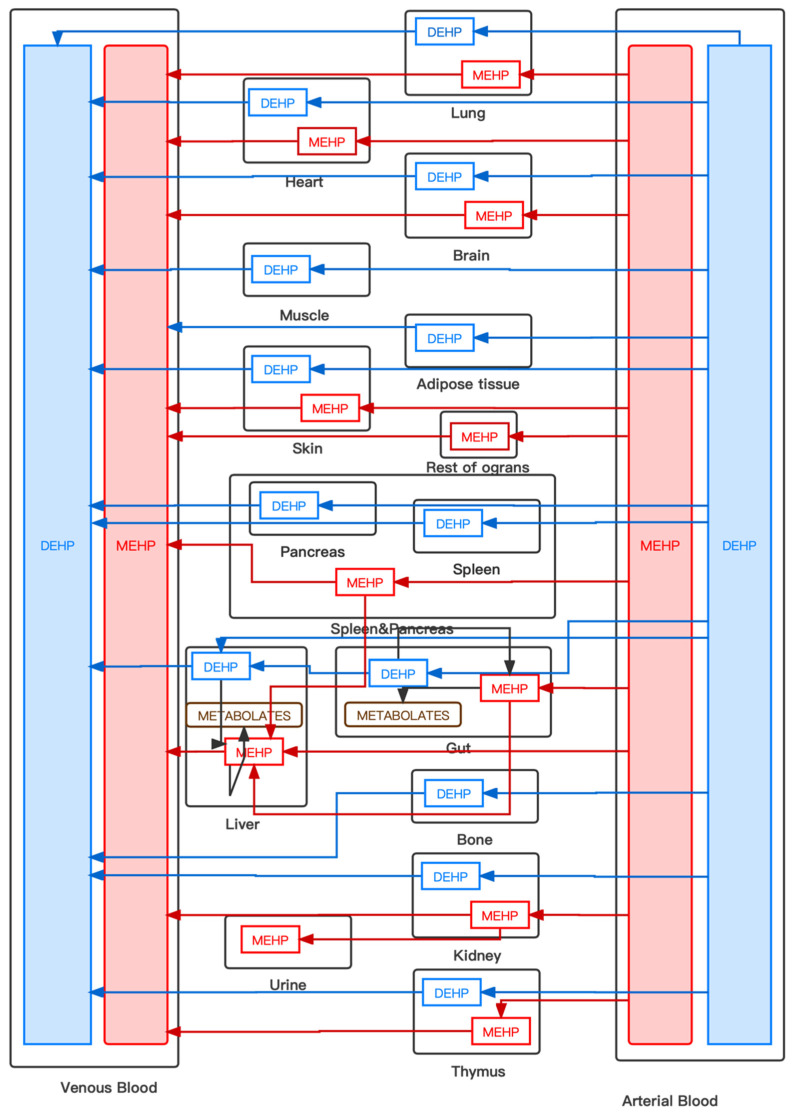
The framework of the PBPK model.

**Figure 2 ijerph-19-05742-f002:**
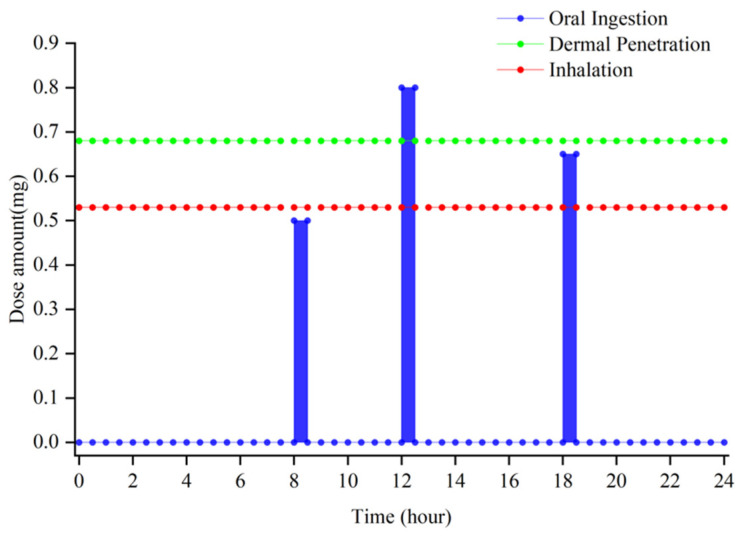
The amount of the three-dose set in the PBPK model.

**Figure 3 ijerph-19-05742-f003:**
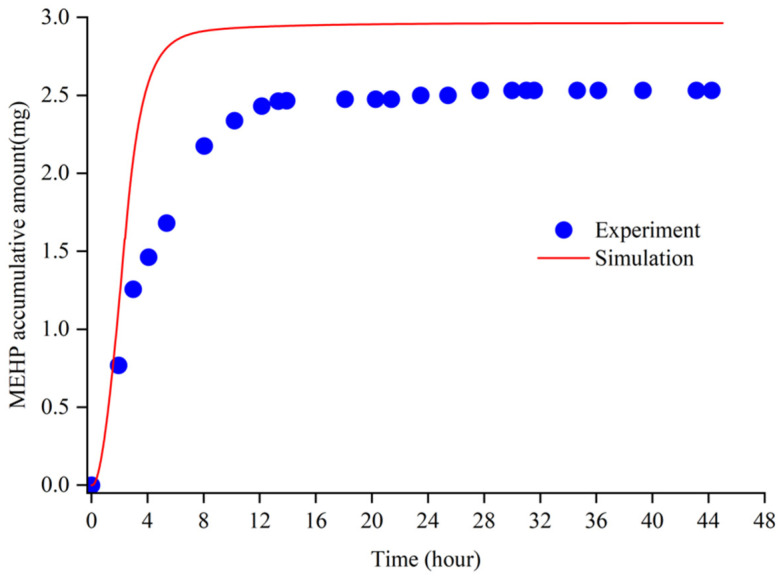
The MEHP accumulative excretion in the urine between the experiment and the simulation.

**Figure 4 ijerph-19-05742-f004:**
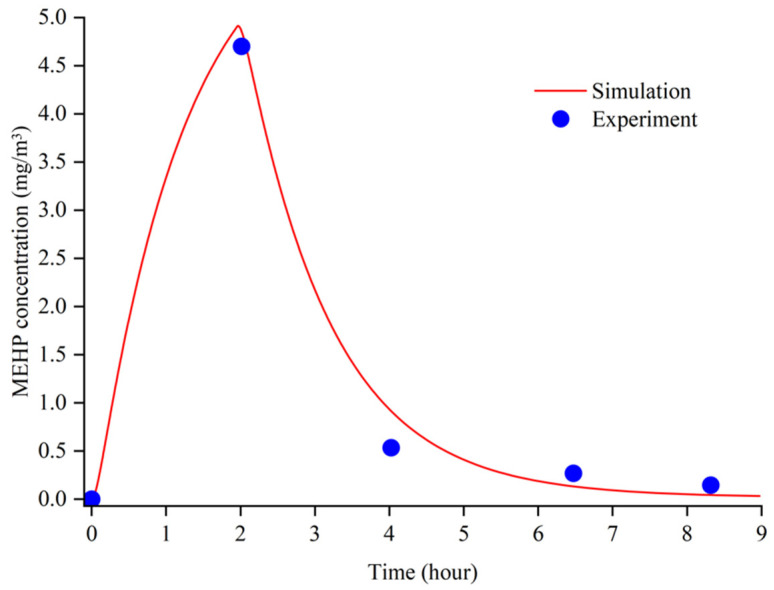
The MEHP concentration in the plasma between the experiment and the simulation.

**Figure 5 ijerph-19-05742-f005:**
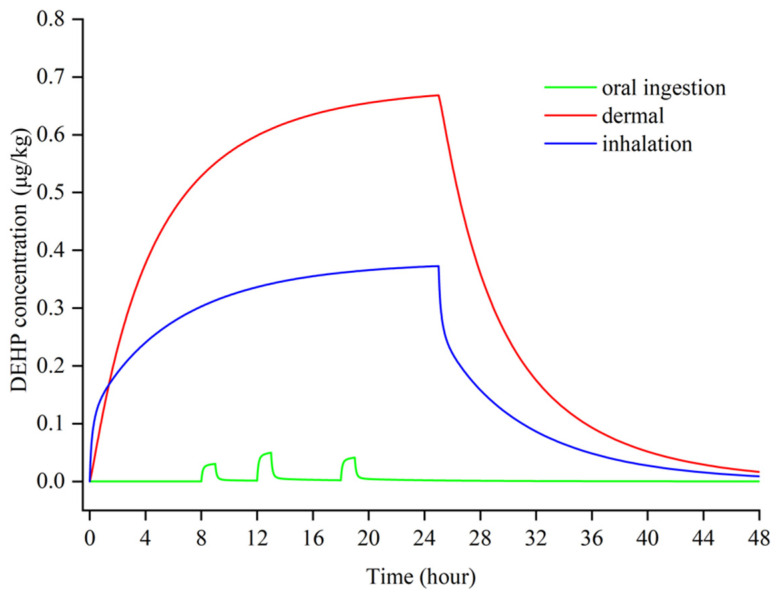
The total accumulation through three intake methods under 24 h doses.

**Figure 6 ijerph-19-05742-f006:**
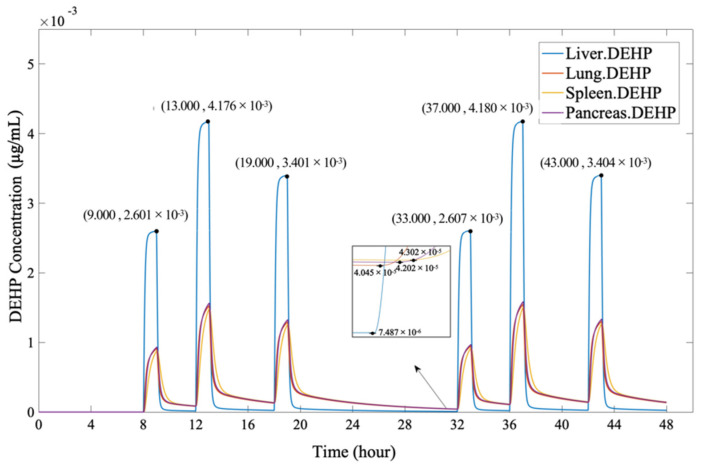
The DEHP concentration in organs, through oral ingestion.

**Figure 7 ijerph-19-05742-f007:**
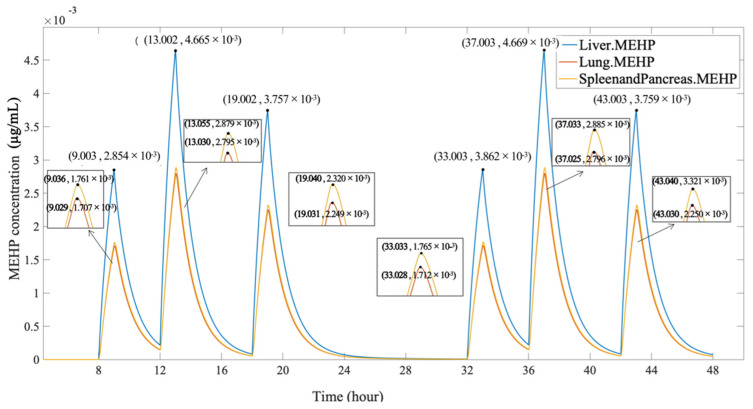
The MEHP concentration in the organs, through oral ingestion.

**Figure 8 ijerph-19-05742-f008:**
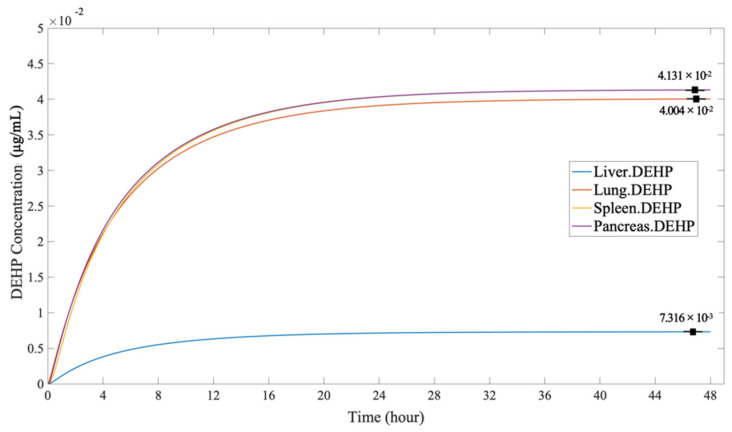
The DEHP concentration in the organs, through dermal penetration.

**Figure 9 ijerph-19-05742-f009:**
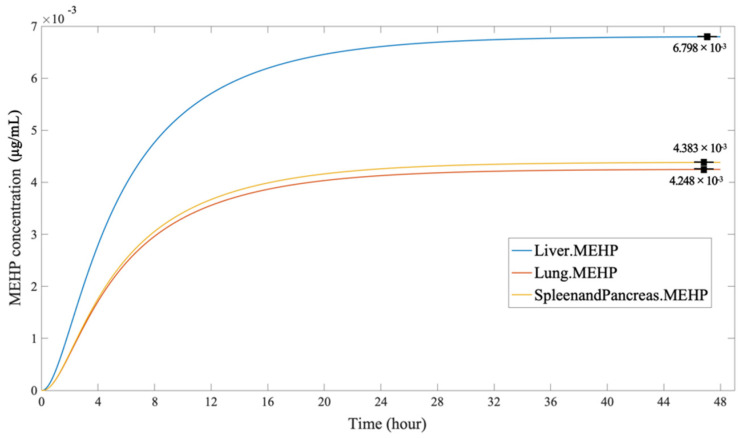
The MEHP concentration in the organs, through dermal penetration.

**Figure 10 ijerph-19-05742-f010:**
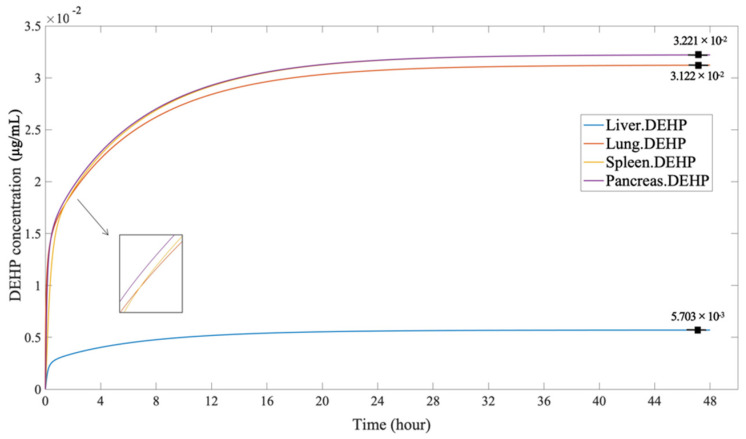
The DEHP concentration in the organs, through inhalation.

**Figure 11 ijerph-19-05742-f011:**
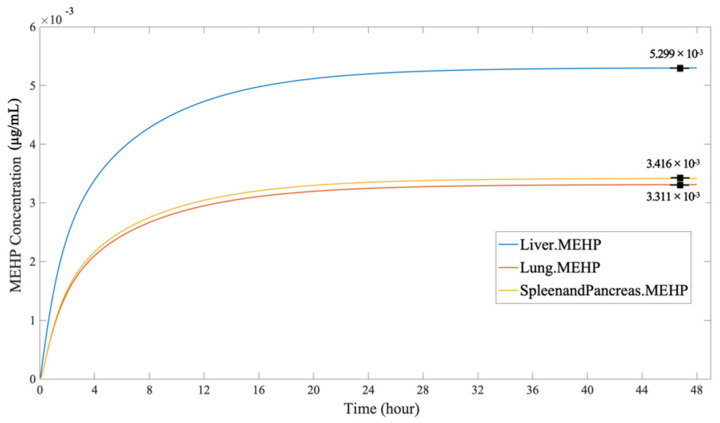
The MEHP concentration in the organs, through inhalation.

**Figure 12 ijerph-19-05742-f012:**
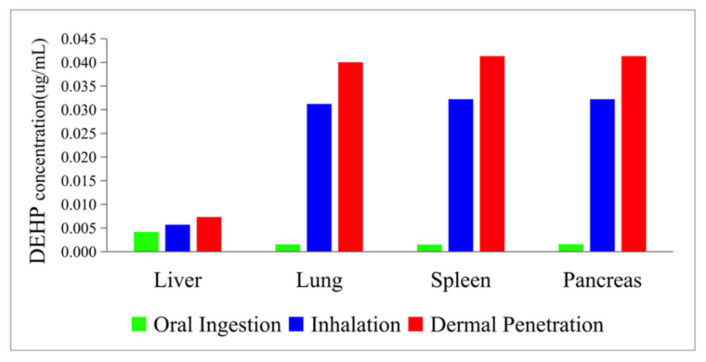
The maximum DEHP concentration in the organs.

**Figure 13 ijerph-19-05742-f013:**
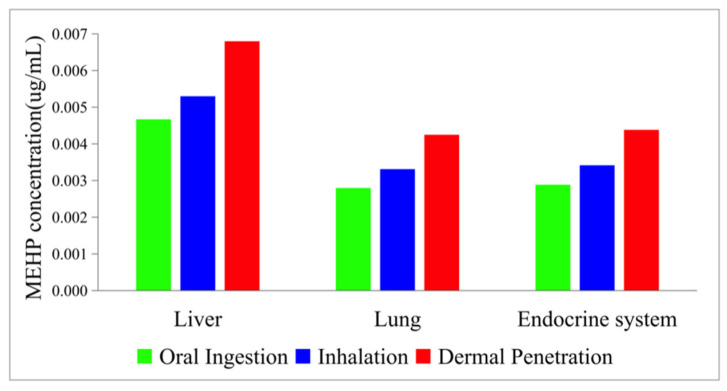
The maximum MEHP concentration in the organs.

**Table 1 ijerph-19-05742-t001:** The PBPK model parameter.

Parameters	Symbol	Unit	Values/ Contribution	Reference
Molecular weight (DEHP)	MW	g/mole	391	-
Molecular weight (D4-MEHP)	MW	g/mole	281	[28]
Molecular weight (MEHP-OH)	MW	g/mole	297	[28]
Molecular weight (D4-5-oxo MEHP)	MW	g/mole	295	[28]
Molecular weight (D4-5-cx MEPP)	MW	g/mole	311	[28]
Octanol: water partition coefficient	LogKo:w	N	7.6	-
Partition coefficients				
Gut/plasma	k_gut_plasma	N	LN ^a^ (12.86,1.1)	-
Liver/plasma	k_liver_plasma	N	LN ^a^ (10.16,1.1)	-
Fat/plasma	k_fat_plasma	N	LN ^a^ (188, 1.1)	-
Rest of the body/plasma	k_restbody_plasma	N	LN ^a^ (6.24, 1.1)	-
Liver/plasma	k_liver_plasmaM1	N	LN ^a^ (1.7, 1.1)	[32]
Fat/plasma	k_fat_plasmaM1	N	LN ^a^ (0.12, 1.1)	[32]
Rest of the body/plasma	k_restbody_plasmaM1	N	LN ^a^ (0.38, 1.1)	[33]
Uptake rate of 5-OHMEHP to blood	KtM2	1/h	LN ^a^ (0.07, 1.5)	[34]
Uptake rate of 5-oxo MEHP to blood	KtM4 cytosol maximum reaction value	1/h	LN ^a^ (0.08, 1.5)	[34]
Absorption and elimination parameters				
Unbound fraction in plasma for MEHP	fup	N	0.007	[35]
Oral absorption rate	kgut	1/h	LN ^a^ (7, 1.5)	[35]
Elimination constant	kurine	1/h	LN ^a^ (0.35, 1.1)	[27]
Metabolic parameters for DEHP and its metabolites in the intestines and the liver				
DEHP to MEHP in intestinal MSP ^b^ maximum reaction value	vmaxgutM1	μg/min/mg MSP ^b^	LN ^a^ (0.11,1.1) ^c^	[36]
DEHP to MEHP in gut cytosol MSP ^b^ maximum reaction value	vmaxgut_cytM1	μg/min/mg	LN ^a^ (0.312,1.1) ^c^	[36]
MEHP to 5-OH MEHP maximum reaction value	vmaxgutM2	μg/min/mg MSP ^b^	LN ^a^ (0.0012,1.1) ^c^	[36]
MEHP to 5-carboxy MEPP maximum reaction value	vmaxgutM3	μg/min/mg MSP ^b^	0	[36]
MEHP-OH to 5-oxo MEHP maximum reaction value	vmaxgutM4	μg/min/mg MSP ^b^	LN ^a^ (0.0012,1.5) ^c^	[36]
MEHP to phthalic acid esters maximum reaction value	vmaxgutM5	mg/min/mg MSP ^b^	LN ^a^ (0.285, 1.1) ^c^	[36]
Conc. at half maximum value	kmgutM1	μg/L	6956	[36]
Conc. at half maximum value	kmgutM2	μg/L	22508	[36]
Conc. at half maximum value	kmgutM3	μg/L	0	[36]
Conc. at half maximum value	kmgutM4	μg/L	219076	[36]
Conc. at half maximum value	kmgutM5	μg/L	187652	[36]
Conc. at half maximum value	kmgut_cytM1	μg/L	7038	[36]
DEHP to MEHP in liver MSP maximum reaction value	vmaxliverM1	μg/min/mg MSP ^b^	LN ^a^ (0.112, 1.1) ^c^	[36]
DEHP to MEHP in liver cytosol maximum reaction value	vmaxliverM1_cyt	μg/min/mg	LN ^a^ (0.036, 1.1) ^c^	[36]
MEHP to 5-OH MEHP maximum reaction value	vmaxliverM2	μg/min/mg MSP ^b^	LN ^a^ (0.172, 1.1) ^c^	[36]
MEHP to 5-carboxy MEPP maximum reaction value	vmaxlivM3	μg/min/mg MSP ^b^	LN ^a^ (0.0023, 1.5) ^c^	[36]
MEHP-OH to 5-oxo MEHP maximum reaction value	vmaxlivM4	μg/min/mg MSP ^b^	LN ^a^ (0.003, 1.1) ^c^	[36]
MEHP to phthalic acid esters maximum reaction value	vmaxlivM5	μg/min/mg MSP ^b^	LN ^a^ (0.088, 1.1) ^c^	[36]
Conc. at half maximum value	kmliver_cytM1	μg/L	2228.7	[36]
Conc. at half maximum value	kmliverM2	μg/L	7980.4	[36]
Conc. at half maximum value	kmliverM3	μg/L	1124	[36]
Conc. at half maximum value	kmliverM4	μg/L	23117.7	[36]
Conc. at half maximum value	kmliverM5	μg/L	141315	[36]

^a^ LN represents that the model parameters are distributed log normally in the range of ±1 to ±1.5 standard deviations. ^b^ MSP represents the macrophage stimulating protein. ^c^ The parameter value needs to be scaled to the whole body weight prior to the model.

**Table 2 ijerph-19-05742-t002:** The organs’ capacity [30].

Compartment	Unit	Value
Arterial Blood	milliliter	1698
Bone	milliliter	4579
Brain	milliliter	1450
Gut	milliliter	1650
Heart	milliliter	310
Kidney	milliliter	280
Liver	milliliter	1690
Lungs	milliliter	1172
Muscle	milliliter	35,000
Pancreas	milliliter	77
Rest other organs	milliliter	49,579
Skin	milliliter	7800
Spleen	milliliter	192
Thymus	milliliter	29
Urine	milliliter	1
Venous Blood	milliliter	3396

**Table 3 ijerph-19-05742-t003:** The difference between each day.

	δ (24 h–48h) (%)	δ (48 h–72 h) (%)	δ (72 h–96 h) (%)	δ (96 h–120 h) (%)
Oral ingestion	2.0044	0.0674	0.0023	0.0001
Inhalation	2.0869	0.0700	0.0024	0.0001
Dermal penetration	1.7294	0.0979	0.0032	0.0002

## Data Availability

Adapted with permission from ref. [30]. 2008 Peters, S.A.

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
