# Peer review of "Modeling di (2-ethylhexyl) Phthalate (DEHP) and Its Metabolism in a Body’s Organs and Tissues through Different Intake Pathways into Human Body"

_ijerph, 2022, doi:10.3390/ijerph19095742_

Round 1
Reviewer 1 Report
The authors constructed an improved PBPK (Physiologically Based Pharmacokinetic) model to simulate the DEHP metabolism. The model was validated with the data of humans orally exposed to DEHP. A set of oral, dermal, and inhalation doses were given for the simulation. The obtained results suggest the importance of dermal exposure. The model may be useful for understanding the DEHP metabolism in the body.
Major points
- Lines 117-118, How exactly ‘a new dosing amount of three kinds of intake‘ (Fig.2) was determined. Please explain it logically/quantitatively ‘based on studies above and a field investigation of health exposure to DEHP‘.
- In Fig.4, the simulated line is not fit with experiment data well. In the simulation, the level reached the max in 4-5 h, vs. the actual data took 12 h to reach its max. It means some parameters in the model need to be modified for optimization. Again, a rational explanation is needed to determine if this validates the model.
Minor points
- Lines 180, 192, etc. Please correct Figure numbers. They should be Figure 7-1, Figure 7-2, and so on.
- Please correct the English format throughout the manuscript. For instance, spacing should be kept in order before parenthesis or between words.
Reviewer 2 Report
The manuscript by Li et al. use physiologically based pharmacokinetic (PBPK) modeling to report on the differing rates of phthalate uptake, distribution and elimination subject to the various routes of exposure, such as by inhalation, ingestion or dermal exposure. This work can have benefits with prioritizing exposure risk assessments and can guide body-burden analysis to prioritize exposure assessments. Overall the paper is well written with minor comments listed below:
Line 30: change "..and increase exposure to.." to "..that can increase exposure to.."
Line 32: start the sentence as: "Epidemiological and toxicological ....shown that exposure to PAEs through indoor sources may be..."
Line 39: change to "..on the endocrine system.."
Line 45: change to "..with numerous measured values.."
Line 72: I am not sure why reference [28] is being cited to refer to the 14-compartment PBPK model in Matlab. The work of Bornehag et al., (2004) makes no mention of using such a PBPK model.
Lines 13 and 85: The full name for each chemical should be given at the first instance of mentioning DEHP and MEHP (i.e., di-ethylhexyl phthalate, etc.)
Lines 92 - 93: same as above - full names should be given for each acronym.
Lines 92 and 101: The citation of Sharma and Peters should be followed by the reference index.
Line 103: Table 1 title should be more descriptive. What is 'LN', Null should be consistently written with a capital 'N' or lower case 'n', what is M1, M2, MSP, etc.?
Line 122: dinners instead of diners
Line 144: should be "..it needs study.."
Line 163: Figure legend should have units in greek symbol for micro gram instead of ug
Line 227 and 237: Figure 8 should be demarcated as Figure 8-1 and 8-2
Line 244: Unclear sentence, what do you mean by DEHP and MEHP are tested to do damage to human organs?
Line 259 and 262: excessive spacing between figure and figure title.
Reviewer 3 Report
The article is interesting, being an important added value to apply the doses obtained to mathematical models. However, I did not understand where the authors were getting the values? the conditions of the values used were the same? for example figure 3, where was the original data taken from?
However, I consider the article to have good bibliographic support, and the model is well discussed and analysed, and can be an important added value for the interpretation of new data
Reviewer 4 Report
It is a good and practical study but I made the following observations which should be addressed by the authors. After doing the following points, it can be accepted in the journal of ijerph.
- It is recommended that the article be reviewed by an English language expert.
Title:
You should mention “di(2- ethylhexyl) phthalate (DEHP)”.
Abstract:
- Line 7: mention “Phthalate acid esters”, correct the whole text.
- Line 8, 9, 11, etc. in the whole text, use the “PAE” instead of “Phthalate esters” and avoid repeating the whole word.
- Express the minimum and maximum amount of accumulation numerically in the abstract.
- Abstract should include brief introduction, method, results and conclusion.
Keywords:
- The words should be listed in alphabetical order.
Introduction
- There are many writing errors, they must be reviewed by an English expert, such as page2, line 57: you should say “PAEs distribution in body’s organs” or “distribution of phthalate esters in body’s organs” and …
- The innovations of this research and the necessity of doing it should be stated in more detail at the end of the introduction.
- You can use the following articles to complete the introduction section:
https://doi.org/10.1080/03067319.2022.2062239, https://doi.org/10.1016/j.microc.2020.105516
Materials and methods:
- p2, line 85. Define “MEHP”
- section 2-1. Avoid what you said in the introduction in the Materials and Methods section.
- Explain Figure 1 further (in the title of figure), for example, what are the colored lines of the sign and other ambiguities in this figure.
- Table 1. Define “Null”, “MW”, “LN”, “logKo-W”, “K-gut”, “MSP”, “vmaxgut_cytM1”, “fup” and etc. in this table. All summary words must be defined first. You can define these words in a table footer.
-What difference between “null” and “Null”?
- There should be a space between numbers and units, corrections should be made throughout the article.
- Write more and clearer explanations for the title of Figure 2 so that it can express this shape.
- Are materials, devices and tools not used, which should be mentioned in this section with company name and other characteristic??!!
Results:
- Title of this section should be change to “Results and discussion”
-The title of Table 4 is different from what is stated in the text, “MEHP” or “DEHP”??
- Check numbers in this sentence “While the difference 156 between 48h and 72h was 0.0979% to 0.0001%, and the following error was getting smaller 157 gradually.”
- In Figure 5, why does it show 48 hours if it is mentioned in the title 24 hours?
- Title of Figure 6 (both of them) change to “Figure 6-1” and “Figure 6-2”, respectively. And also about Figure 7 and Figure 8.
- P10, line 186. Check this number “Figure 7-2”
- It seems that the discussion part is a weak in all parts. More explanations and reasons should be given, as well as more comparisons with other studies.
Conclusions
- Much of what is mentioned in the concluding section is repetitive. In this section, the result of the work should be clearly stated and the achievements of the article should be fully explained. Finally, possible suggestions should be mentioned.
References
- Check the references section for the journal guide.

Round 2
Reviewer 4 Report
The article can be accepted